# ADVERSARY-AWARE PARTIAL LABEL LEARNING WITH LABEL DISTILLATION

## ABSTRACT

To ensure that the data collected from human subjects is entrusted with a secret, rival labels are introduced to conceal the information provided by the participants on purpose. The corresponding learning task can be formulated as a noisy partial-label learning problem. However, conventional partial-label learning (PLL) methods are still vulnerable to the high ratio of noisy partial labels, especially in a large labelling space. To learn a more robust model, we present Adversary-Aware Partial Label Learning and introduce the *rival*, a set of noisy labels, to the collection of candidate labels for each instance. By introducing the rival label, the predictive distribution of PLL is factorised such that a handy predictive label is achieved with less uncertainty coming from the transition matrix, assuming the rival generation process is known. Nonetheless, the predictive accuracy is still insufficient to produce an sufficiently accurate positive sample set to leverage the clustering effect of the contrastive loss function. Moreover, the inclusion of rivals also brings an inconsistency issue for the classifier and risk function due to the intractability of the transition matrix. Consequently, an adversarial teacher within momentum (ATM) disambiguation algorithm is proposed to cope with the situation, allowing us to obtain a provably consistent classifier and risk function. In addition, our method has shown high resiliency to the choice of the label noise transition matrix. Extensive experiments demonstrate that our method achieves promising results on the CIFAR10, CIFAR100 and CUB200 datasets.

## 1 INTRODUCTION

Deep learning algorithms depend heavily on a large-scale, true annotated training dataset. Nonetheless, the costs of accurately annotating a large volume of true labels to the instances are exorbitant, not to mention the time invested in the labelling procedures. As a result, weakly supervised labels such as partial labels that substitute true labels for learning have proliferated and gained massive popularity in recent years. Partial-label learning (PLL) is a special weakly-supervised learning problem associated with a set of candidate labels $\vec{Y}$ for each instance, in which only one true latent label $y$ is in existence. Nonetheless, without an appropriately designed learning algorithm, the limitations of the partial label are evident since deep neural networks are still vulnerable to the ambiguous issue rooted in the partial label problem because of noisy labels Zhou (2018); Patrini et al. (2017); Han et al. (2018). As a result, there have had many partial label learning works (PLL)Cour et al. (2011); Hüllermeier & Beringer (2006); Feng & An (2019); Feng et al. (2020) successfully solved the ambiguity problem where there is a set of candidate labels for each instance, and only a true label exists. Apart from the general partial label, we have also seen a variety of partial label generations evolved, simulating different real-life scenarios. The independently and uniformly drawing is the one have seen the most Lv et al. (2020); Feng & An (2019). The other problem settings include the instance dependent partial label learning, where each partial label set is generated depending on the instance as well as the true label Xu et al. (2021). Furthermore, Lv et al. (2020) has introduced label specific partial label learning, where the uniform flipping probability of similar instances differs from dissimilar group instances. Overall, the learning objective of the previous works is all about disambiguation. More specifically, the goal is to design a classifier training with partial labels, aiming to correctly label the testing dataset, hoping the classification performance will be as close as the full supervised learning.

On the contrary, there is a lack of discussion on previous works that shed light on the data privacy-enhancing techniques in general partial label learning. The privacy risk is inescapable; thus, privacy-preserving techniques need to be urgently addressed. Recently, we have seen surging data breach cases worldwide. These potential risks posed by the attacker are often overlooked and pose a detrimental threat to society. For instance, it is most likely for the adversary to learn from stolen or leaked partially labelled data for illegal conduct using the previous proposed partial-label learning methods. Subsequently, it has become an inherent privacy concerns in conventional partial label learning. In this paper, the Adversary-Aware partial label learning is proposed to address and mitigate the ramification of the data breach. In a nutshell, we propose an affordable and practical approach to manually corrupt the collected dataset to prevent the adversary from obtaining high-quality, confidential information meanwhile ensure the trustee has full access to the useful information. However, we have observed that adversary-aware partial label learning possesses some intrinsic learnability issues. Firstly, the intractability is raised from the transition matrix. Secondly, the classifier and risk inconsistency problem has been raised. Hence, we propose an the Adversarial teacher within momentum (ATM)(In section 2.1), adversary-aware loss function equation 19, and a new ambiguity condition equation 1 to counter the issues.

Under the adversary-aware partial label problem setting, the rival is added to a candidate set of labels. To achieve that, we extend the original partial label generation equation 2 by factorisation to add the rival $Y'$. Subsequently, we have the adversary-aware partial label generation established as equation 3. Then, we decompose the second equation of equation 3 into the rival embedded intractable transition matrix term $Q^*$ and class instance-dependent transition matrix $T_{y,y'}$, which is $P(Y' = y' \mid Y = y, X = x)$. In our problem setting, $\overline{T}_{y,y'}$, the class instance-independent transition matrix is utilised, which is defined as $P(Y' = y' \mid Y = y)$, with the assumption the rival is generated depending only on $Y$ but instance $X$. Under the assumption, the class instance-independent transition matrix is simplified and mathematically identifiable. Since all the instances share the same class instance-independent transition matrix in practice, such encryption is more affordable to implement. The rival variable serves as controllable randomness to enhance privacy against the potential adversary and information leakage. In contrast, the previous methods can not guarantee the privacy protection property.

However, a fundamental problem has been raised, inclusion of the rival implies an inconsistent classifier according to the adversary-aware label generation equation equation 3. Learning a consistent partial label classifier is vital, but in our problem setting, the consistency classifier may not be obtained due to the intractability of $Q^*$(details are described in section 1.2). As a consequence, the Adversarial teacher within momentum (ATM) is proposed, which is designed to identify the term $P(\vec{Y} \mid Y, Y', X)$ which is denoted as $Q^*$. The Moco-style dictionary technique He et al. (2020) and Wang et al. (2022) have inspired us to explore exploiting the the soft label from instance embedding, leveraging $\overline{T}_{y,y'}$ to identify or reduce the uncertainty of the $Q^*$ due to the property of informational preservation and tractability. Therefore, a consistent partial label learner is obtained if the uncertainty raised from the transition matrix is reduced greatly. Specifically, we transform the inference of label generation in Adversary-Aware PLL as an approximation for the transition matrix $Q^*$. Ultimately, a tractable solution to the unbiased estimate of $P(\vec{Y} \mid Y, Y', X)$ can be derived. Lastly, we have rigorously proven that a consistent Adversary-Aware PLL classifier can be obtained if $P(\vec{Y} \mid Y, Y', X)$ and $P(Y' \mid Y)$ are approximated accurately according to equation 3.

In this work, we are mainly focusing on identifying the transition matrix term $P(\vec{Y} \mid Y, Y', X)$. The rival is generated manually for privacy enhancement. Thus the $P(Y' \mid Y)$ is given by design. Overall, our proposed method has not only solved the ambiguity problem in Adversary-Aware PLL but also addressed the potential risks from the data breach by using a rival as the encryption. Our proposed label generation bears some resemblance to local differential privacy Kairouz et al. (2014); Warner (1965), which aims to randomise the responses. The potential application is to randomise survey responses, a survey technique for improving the reliability of responses to confidential interviews or private questions. Depending on the sophistication of the adversary, our method offers a dynamic mechanism for privacy encryption that is more resilient and flexible to face the potential adversary or privacy risk. By learning from the previous attacks, we can design different levels of protection by adjusting the $\overline{T}$ term. The **main contributions** of the work are summarized:

- We propose a novel problem setting named adversary-aware partial label learning.

- We propose a novel Adversary-Aware loss function and the Adversarial teacher within momentum (ATM) disambiguation algorithm. Our proposed paradigm and loss function can be applied universally to other related partial label learning methods to enhance the privacy protection.
- A new ambiguity condition (equation 1) for Adversary-Aware Partial Label Learning is derived. Theoretically, we proven that the method is a Classifier-Consistent Risk Estimator.

## 1.1 RELATED WORK

**Partial Label Learning (PLL)** trains an instance associated with a candidate set of labels in which the true label is included. Many frameworks are designed and proposed to solve the label ambiguity issue in partial label learning. The probabilistic graphical model-based methodsZhang et al. (2016); Wang & Isola (2020); Xu et al. (2019); Lyu et al. (2019) as well as the clustering-based or unsupervised approaches Liu & Dietterich (2012) are proposed by leveraging the graph structure and prior information of feature space to do the label disambiguation. The average-based perspective methods Hüllermeier & Beringer (2006); Cour et al. (2011); Zhang et al. (2016) are designed based on the assumption of uniform treatment of all candidates; however, it is vulnerable to the false positive label, leading to misled prediction. Identification perspective-based methods Jin & Ghahramani (2002) tackle disambiguation by treating the true label as a latent variable. The representative perspective approach uses the maximum margin method Nguyen & Caruana (2008); Wang et al. (2020; 2022) to do the label disambiguation. Most recently, self-training perspective methodsFeng & An (2019); Wen et al. (2021); Feng et al. (2020) have emerged and shown promising performance. In **Contrastive Learning** He et al. (2020); Oord et al. (2018), the augmented input is applied to learns from feature of the unlabeled sample data. The learning objective is to differentiate the similar and dissimilar parts of the input, in turn, maximise the learning of the high-quality representations. CL has been studied in unsupervised representation fashion Chen et al. (2020); He et al. (2020), which treats the same classes as the positive set to boost the performance. The weakly supervised learning has also borrowed the concepts of CL to tackle the partial label problem Wang et al. (2022). The CL has also been applied to semi-supervised learning Li et al. (2020).

## 1.2 ADVERSARY-AWARE PARTIAL LABEL PROBLEM SETTING

Given the input space $\mathcal{X} \in \mathbb{R}^d$ and label space is defined as $\mathcal{Y} = [c] \in \{1 \cdots c\}$ with the number of $c > 2$ classes. Under adversary-aware partial labels, each instance $X \in \mathcal{X}$ has a candidate set of adversary-aware partial labels $\vec{Y} \in \vec{\mathcal{Y}}$. The adversary-aware partial label set has space of $\vec{\mathcal{Y}} := \{\vec{y} \mid \vec{y} \subset \mathcal{Y}\} = 2^{[c]}$, in which there is total $2^{[c]}$ selection of subsets in $[c]$. The objective is to learn a classifier with the adversary-aware partially labelled sample $n$, which was i.i.d drawn from the $\vec{\mathcal{D}} = \{(X_1, \vec{Y}_1), \ldots, (X_n, \vec{Y}_n)\}$, aiming that it is able to assign the true labels for the testing dataset. Given instance and the adversary-aware partial label $\vec{Y}$ the adversary-aware partial label dataset distribution $\vec{D}$ is defined as $(X, \vec{Y}) \in \mathcal{X} \times \vec{\mathcal{Y}}$. The class instance-independent transition matrix $P(Y' \mid Y)$ is denoted as $\bar{T} \in \mathbb{R}^{c \times c}$. $\bar{T}_{y,y'} = P(Y' = y' \mid Y = y)$ where $\bar{T}_{y,y} = 0, \forall y', y \in [c]$. The adversary-aware means the designed paradigm can prevent the adversary from efficiently and reliably inferring certain information from the database without the $\bar{T}$, even if the data was leaked. The rival is the controllable randomness added to the partial label set to enhance privacy.

### 1.2.1 ASSERTION CONDITIONS IN LABEL GENERATION SET

The following conditions describe the learning condition for adversary-aware partial label. According to Cour et al. (2011) there needs to be certain degrees of ambiguity for the partial label learning. Lemma 1 is the new ERM learnability condition which is proposed as follows

$$P_{y',\bar{y}} := P(y', \bar{y} \in \vec{Y} \mid Y' = y', \bar{Y} = \bar{y}, X = x). \tag{1}$$

The $y'$ is the rival, and $\bar{y}$ is the false positive label that exists in the partial label set. It has to be met to ensure the Adversary-Aware PLL problem is learnable with $y' \neq y$ and $\bar{y} \neq y$, these conditions ensure the ERM learnability Liu & Dietterich (2014) of the adversary-aware PLL problem if there is small ambiguity degree condition. In our case which is that, $P_{y',\bar{y}} < 1$. The $y$ is the true label corresponding to each instance $x$. And $P_y := P(y \in \vec{Y} \mid Y = y, X = x)$, where $P_y = 1$ to ensure that the ground truth label is in the partial label set with respect to each instance.

### 1.2.2 LABEL GENERATION

In the previous works of partial label generation procedure, only a candidate of the partial label was generated as such.

**The Standard Partial Label Generation:**

$$
\sum_{y \in Y} \mathrm{P}(\vec{Y} = \vec{y}, Y = y \mid X = x) = \sum_{y \in Y} \mathrm{P}(\vec{Y} = \vec{y} \mid Y = y, X = x)\mathrm{P}(Y = y \mid X = x).
$$
$$
= \sum_{y \in Y} \mathrm{P}(\vec{Y} = \vec{y} \mid Y = y)\mathrm{P}(Y = y \mid X = x),
$$
(2)

where $\mathrm{P}(\vec{Y} = \vec{y} \mid Y = y, X = x)$ is the label generation for the class instance-dependent partial label and $\mathrm{P}(\vec{Y} = \vec{y} \mid Y = y)$ is the standard partial label learning framework. Then we present the difference between the general partial labels and the adversary-aware partial label.

**The Adversary-Aware Partial Label Generation:**

$$
\sum_{y \in Y} \mathrm{P}(\vec{Y} = \vec{y} \mid X = x) = \sum_{y \in Y} \sum_{y' \in Y'} \mathrm{P}(\vec{Y} = \vec{y}, Y = y, Y' = y' \mid X = x)
$$
$$
= \sum_{y \in Y} \sum_{y' \in Y'} \underbrace{\mathrm{P}(\vec{Y} = \vec{y} \mid Y = y, Y' = y', X = x)}_{\textbf{Adversary-Aware transition matrix}} \bar{T}_{y,y'}\mathrm{P}(Y = y \mid X = x).
$$
(3)

In the adversary-aware partial label problem setting, the transition matrix of the adversary-aware partial label is defined as $\mathrm{P}(\vec{Y} \mid Y, Y', X)$ and denoted as $Q^* \in \mathbb{R}^{c \times (2^c - 2)}$. The partial label transition matrix $\mathrm{P}(\vec{Y} \mid Y)$ is denotes as $\bar{Q} \in \mathbb{R}^{c \times (2^c - 2)}$. Theoretically, if the true label $Y$ of the vector $\vec{Y}$ is unknown given an instance $X$, where $\vec{y} \in \vec{Y}$ and there are $2^c - 2$ candidate label sets. The $\epsilon_x$ is the instance-dependent rival label noise for each instance where $\epsilon_x \in \mathbb{R}^{1 \times c}$. The entries of the adversary-aware transition matrix for each instance is defined as follows

$$
\sum_{j=1}^{2^c-2} Q^*[:, j] = \sum_{j=1}^{2^c-2} ([\bar{Q}[:, j]^T + \epsilon_x]\bar{T})^T = \sum_{j=1}^{2^c-2} (A[:, j]^T \bar{T})^T,
$$
(4)

where $A[:, j]^T = \bar{Q}[:, j]^T + \epsilon_x$ and the conditional distribution of the adversary-aware partial label set $\vec{Y}$ based on Wen et al. (2021) is derived as belows

$$
\mathrm{P}(\vec{Y} = \vec{y} \mid Y = y, Y' = y', X = x) = \prod_{b' \in \vec{y}, b' \neq y} p_{b'} \cdot \prod_{t' \notin \vec{y}} (1 - p_{t'}),
$$
(5)

where $p_{t'}$ and $p_{b'}$ are defined as

$$
p_{t'} := \mathrm{P}(t \in \vec{Y} \mid Y = y, Y' = y', X = x) < 1, p_{b'} := \mathrm{P}(b \in \vec{Y} \mid Y = y, Y' = y', X = x) < 1.
$$
(6)

We summarize the equation 3 as a matrix form in equation 7. The inverse problem is to identify a sparse approximation matrix $\boldsymbol{A}$ to use equation 8 to estimate the true posterior probability.

$$
\underbrace{P(\vec{Y} \mid X = x)}_{\textbf{Adversary-aware PLL}} = \boldsymbol{Q^*} \underbrace{P(Y \mid X = x)}_{\textbf{True posterior probability}},
$$
$$
\boldsymbol{Q^*}^{-1} \underbrace{P(\vec{Y} \mid X = x)}_{\textbf{Adversary-aware PLL}} = \underbrace{P(Y \mid X = x)}_{\textbf{True posterior probability}},
$$
(7)

$$
\bar{\boldsymbol{T}}^{-1}\boldsymbol{A}^{-1} \underbrace{P(\vec{Y} \mid X = x)}_{\textbf{Adversary-aware PLL}} \approx \underbrace{P(Y \mid X = x)}_{\textbf{True posterior probability}}.
$$
(8)

In reality, due to the computational complexity of the transition matrix, it would be a huge burden to estimate $Q^*$ accurately for each instance. The $2^c - 2$ is an extremely large figure and increases

exponentially as the label space increase. Therefore, we are no longer required to estimate the true transition matrix $P(\vec{Y} \mid Y, Y', X)$. Instead, we resort to using instance embedding in the form of a soft label to identify the adversary-aware partial label transition matrix $Q^*$. Specifically, we proposed to use a soft pseudo label from the instance embedding (Prototype) to approximate the adversary-aware transition matrix for each instance. The reason is that we can not achieve the true transition matrix $Q^*$ directly due to the nature of the practical partial label problem. Therefore, we have used the self-attention prototype learning to approximate the true transition matrix. The detail is described in section 2.1. Since the Adversary-aware partial label is influenced by the rival label noise, it is challenging to accurately estimate both the class instance-independent transition matrix $\bar{T}$ and the sparse matrix $A$ simultaneously to estimate the true posterior. Considering that the $\bar{T}$ is private and given, it is easier for us just to approximate $A$ to estimate the posterior probability than the adversary. The equation 8 is implemented as the loss function in equation 17.

## 1.3 Positive Sample Set

The construction of a positive sample is used for contrastive learning to identify the transition matrix $P(\vec{Y} \mid Y', Y, X)$ via the label disambiguation. Nonetheless, the performance of the contrastive learning erodes drastically due to the introduced rival, which is manifest in the poorly constructed positive sample set, resulting in the degenerated classification performance (See Figure 2). Subsequently, the adversary-aware loss function is proposed in conjunction the contrastive learning to prevent classification performance degeneration. To start with, we define $L_2$ norm embedding of $u$ and $k$ as the query and key latent feature from the feature extraction network $f_\Theta$ and key neural network $f'_\Theta$ respectively. Correspondingly, we have the output $u \in R^{1 \times d}$ where $u_i = f_\Theta(\mathrm{Aug}_q(x))$ and $z \in R^{1 \times d}$ where $z_i = f'_\Theta(\mathrm{Aug}_k(x_i))$. The construction of a positive sample set is shown as follows. In each mini-batch, we have $\vec{D}_b$ where $\vec{D}_b \in \vec{D}$. The $f(x_i)$ is the function of a neural network with a projection head of 128 feature dimensionality. The outputs of $D_q$ and $D_k$ are defined as follows,

$$D_q = \{u_i = f\left(\mathrm{Aug}_q\left(x_i\right)\right) \mid x_i \in \vec{D}_b\}, \tag{9}$$

$$D_k = \{z_i = f'\left(\mathrm{Aug}_k\left(x_i\right)\right) \mid x_i \in \vec{D}_b\}, \tag{10}$$

where $\bar{S}(x)$ is the sample set excluding the query set $q$ and is defined as $\bar{S}(x) = \bar{C} \backslash \{q\}$, in which $\bar{C} = D_q \cup D_k \cup$ queue . The $D_q$ and $D_k$ are vectorial embedding with respect to the query and key views given the current mini-batch. The queue size is determined accordingly depending on the input. The instances from the current mini-batch with the prediction label $\bar{y}'$ equal to $(\hat{y}_i = c)$ from the $\bar{S}(x)$. is chosen to be the positive sample set. Ultimately, the $N(x)$ is acquired, and it is denoted as

$$N_+(x_i) = \left\{z' \mid z' \in \bar{S}\left(x_i\right), \bar{y}' = (\hat{y}_i = c)\right\}. \tag{11}$$

The $N_+(x)$ is the positive sample set. The construction of sufficiently accurate positive sample set $N_+(x)$ is vital as it underpins the clustering effect of the latent embedding in the contrastive learning procedure. The quality of the clustering effect relies on the precision of prototype $v_j$ corresponding to $j \in \{1, ..., C\}$. Our method helps maintain the precision of prototypes using the $\bar{T}$ to render better label disambiguation module performance for contrastive learning when introduced the rival. where the query embedding $u$ multiplies the key embedding $z$ and then divides with the remaining pool $\bar{C}$. Overall, the $S_+(x)$ is used to facilitate the representation learning of the contrastive learning and the self-attention prototype learning to do the label disambiguation or a more accurate pseudo-labelling procedure. Our proposed loss ensures the prototype and contrastive learning are working systematically and benefit mutually when the rival is introduced. The pseudo label generation is according to equation 16. We have followed Wang et al. (2022) for the positive sample selection.

## 2 Methodology

The main task of partial label learning is label disambiguation, which targets identifying the true label among candidate label sets. Thus, we present an adversarial teacher within momentum (ATM). The equation 17 is developed to do the debiasing from the prediction of $f(x)$ given the adversary-aware partial label via the class instance dependent transition matrix $\bar{T} + I$. The unbiased prediction induces the identification of a more accurate positive sample set which allows Equation 18 to

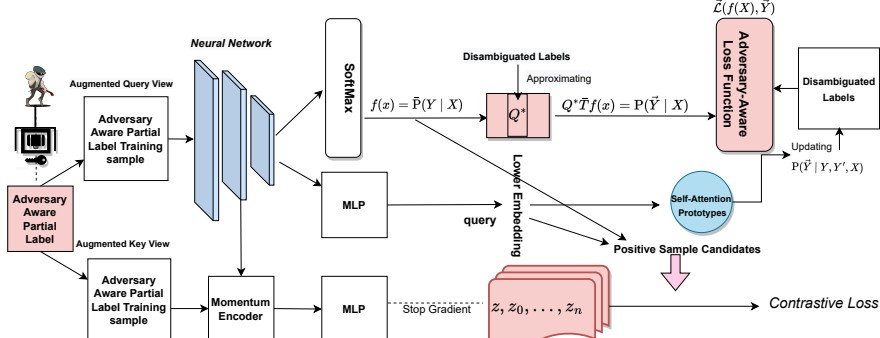

Figure 1: An overview of the proposed method. General partial label can be disclosed to adversary. The initial training is about positive sample selection. Moreover, we have assumed $\bar{T}$ is given.

leverage the high-quality presentation power of a positive sample set to improve the classification performance.

## 2.1 PSEUDO LABEL LEARNERS VIA ADVERSARIAL TEACHER WITHIN MOMENTUM (ATM)

Unlike Wang et al. (2022), we present an adversarial teacher strategy with momentum update (ATM) to guide the learning of pseudo labels using Equation 17. Just like a tough teacher who teaches the subject using harsh contents to test students' understanding of the subject. In our case, the rival is like the subject which is purposely generated by us, at the same time Equation 17 is introduced to check the understanding of the student (classifier) given the scope of testing content which is the $\bar{T}$. Specifically, the spherically margin between prototype vector $\boldsymbol{v}_i \in \mathbb{S}^{d-1}$ and prototype vector $\boldsymbol{v}_j \in \mathbb{S}^{d-1}$ is defined as

$$m_{ij} = \exp\left(-\boldsymbol{v}_i^\top \boldsymbol{v}_j\right). \tag{12}$$

For prototype $\boldsymbol{v}_i$, we define the normalized margin between $\boldsymbol{v}_i$ and $\boldsymbol{v}_j$ as

$$\bar{m}_{ij} = \frac{\exp\left(-\boldsymbol{v}_i^\top \boldsymbol{v}_j\right)}{\sum_{j \neq i} \exp\left(-\boldsymbol{v}_i^\top \boldsymbol{v}_j\right)}. \tag{13}$$

For each $\boldsymbol{v}_i, i \in \{1, \cdots, K\}$, we perform momentum updating with the normalized margin between $\boldsymbol{v}_j$ and $\boldsymbol{v}_i$ for all $j \neq i$ as an regularization. The resulted new update rule is given as

$$\boldsymbol{v}_i^{t+1} = \sqrt{1 - \alpha^2} \boldsymbol{v}_i^t + \alpha \frac{\boldsymbol{g}}{\|\boldsymbol{g}\|_2}, \tag{14}$$

where the gradient $\boldsymbol{g}$ is given as

$$\boldsymbol{g} = \boldsymbol{u} - \beta \sum_{j \neq i} \bar{m}_{ij}^t \boldsymbol{v}_j^t, \tag{15}$$

where $\boldsymbol{u}$ is the query embedding whose prediction is class $i$, $\bar{m}_{ij}^t$ is the normalized margin between prototype vectors at step $t$ (i.e., $\boldsymbol{v}_j^t, j \neq i$). The $v_c$ is the prototype corresponding to each class.

$$\bar{\boldsymbol{q}} = \phi\bar{\boldsymbol{q}} + (1 - \phi)\boldsymbol{v}, \quad v_c = \begin{cases} 1 & \text{if } c = \arg\max_{j \in Y} \boldsymbol{u}^\top \boldsymbol{v} \\ 0 & \text{otherwise}, \end{cases}. \tag{16}$$

where $\bar{q}$ is the target prediction and subsequently used in the equation 17. It was initialised as the uniform probability $\bar{\boldsymbol{q}} = \frac{1}{|c|}\mathbf{1}$ and updated accordingly to the equation 16. The $\phi$ is the hyperparameter controlling for the updating of $\bar{\boldsymbol{q}}$.

## 2.2 ADVERSARY AWARE LOSS FUNCTION.

The goal is to build a risk consistent loss function, hoping it can achieve the same generalization error as the supervised classification risk $R(f)$ with the same classifier $f$. To train the classifier,

we minimize the following modified loss function estimator by leveraging the updated pseudo label from the Adversarial teacher within momentum (ATM) distillation method and transition plus identity matrix, $I_{i,j} \in [0,1]^{c \times c}$, $I_{i,i} = 1$, for $\forall_{i=j} \in [c]$, $I_{i,j} = 0$, for $\forall_{i \neq j} \in [c]$: where $f(\boldsymbol{X}) \in \mathbb{R}^{|c|}$,

$$\vec{\mathcal{L}}(f(X), \vec{Y}) = -\sum_{i=1}^{c} (\bar{q}_i) \log \left( ((\bar{\mathbf{T}} + \mathbf{I})f(X))_i \right).$$

(17)

The proof for the modified loss function is shown in the appendix lemma 4. In our case, given sufficiently accurate positive sample set of the contrastive learning is utilised to incorporate with equation 17 to identify the transition matrix of the adversary-aware partial label. The contrastive loss is defined as follows

$$\mathcal{L}(f(x), \tau, C) = \frac{1}{|D_q|} \sum_{\boldsymbol{u} \in D_q} \left\{ -\frac{1}{N_+(x)} \sum_{\boldsymbol{z}_+ \in N_+(x)} \log \frac{\exp(\boldsymbol{u}^\top \boldsymbol{z}/\tau)}{\sum_{\boldsymbol{z}' \in \bar{C}(\boldsymbol{x})} \exp(\boldsymbol{u}^\top \boldsymbol{z}/\tau)} \right\}.$$

(18)

Finally, we have the Adversary-Aware Loss expressed as

$$\textbf{Adversary-Aware Loss} = \lambda \mathcal{L}(f(x_i), \tau, C) + \vec{\mathcal{L}}(f(X), \vec{Y}).$$

(19)

There are two terms of the proposed loss function (equation 19), which are the equation 17 and equation 18 correspondingly. equation 17 is developed to lessen prediction errors from $f(x)$ given the adversary-aware partial label. The debiasing is achieved via the class instance dependent transition matrix $\bar{T} + I$ by down-weighting the false prediction. The unbiased prediction induces the identification of a more accurate positive sample set. equation 18 is the contrastive loss. It leverages the high-quality representation power of positive sample set to improve the classification performance further.

## 3 THEORETICAL ANALYSIS

The section introduces the concepts of classifier consistency and risk consistency Xia et al. (2019) Zhang (2004), which are crucial in weakly supervised learning. Risk consistency is achieved if the risk function of weak supervised learning is the same as the risk of fully supervised learning with the same hypothesis. The risk consistency implies classifier consistency, meaning classifier trained with partial labels is consistent as the optimal classifier of the fully supervised learning.

**Classifier-Consistent Risk Estimator Learning with True labels.** Lets denote $f(X) = (g_1(x), \ldots, g_K(x))$ as the classifier, in which $g_c(x)$ is the classifier for label $c \in [K]$. The prediction of the classifier $f_c(x)$ is $P(Y = c \mid x)$. We want to obtain a classifier $f(X) = \arg\max_{i \in [K]} g_i(x)$. The loss function is to measure the loss given classifier $f(X)$. To this end, the true risk can be denoted as

$$R(f) = \mathbb{E}_{(X,Y)}[\mathcal{L}(f(X), Y)].$$

(20)

The ultimate goal is to learn the optimal classifier $f^* = \arg\min_{f \in \mathcal{F}} R(f)$ for all loss functions, for instance to enable the empirical risk $\bar{R}_{pn}(f)$ to be converged to true risk $R(h)$. To obtain the optimal classifier, we need to prove that the modified loss function is risk consistent as if it can converge to the true loss function.

**Learning with adversary-aware Partial Label.** An input $X \in \mathcal{X}$ has a candidate set of $\vec{Y} \in \vec{\mathcal{Y}}$ but a only true label $Y \in \vec{\mathcal{Y}}$. Given the adversary-aware partial label $\vec{Y} \in \vec{\mathcal{Y}}$ and instance $X \in \mathcal{X}$ that the objective of the loss function is denoted as

$$\hat{R}(f) = \mathbb{E}_{(X,\vec{Y})} \vec{\mathcal{L}}\left(f(X), \vec{Y}\right).$$

(21)

Since the true adversary-aware partial label distribution $\bar{\mathcal{D}}$ is unknown, our goal is approximate the optimal classifier with sample distribution $\bar{D}_{pn}$ by minimising the empirical risk function, namely

$$\hat{R}_{pn}(f) = \frac{1}{n} \sum_{i=1}^{n} \vec{\mathcal{L}}\left(f(\boldsymbol{x}_i), \vec{y}_i\right).$$

(22)

**Assumption 1.** According to Yu et al. (2018) that the minimization of the expected risk $R(f)$ given clean true population implies that the optimal classifier is able to do the mapping of $f_i^*(X) = P(Y = i \mid X)$, $\forall i \in [c]$. Under the assumption 1, we are able to draw conclusion that $\hat{f}^* = f^*$ applying the theorem 2 in the following.

**Theorem 1.** *Assume that the Adversary-Aware matrix $T_{y,y'}$ is fully ranked and the Assumption 1 is met, the the minimizer of $\hat{f}^*$ of $\hat{R}(f)$ will be converged to $f^*$ of $R(f)$, meaning $\hat{f}^* = f^*$.*

**Remark.** If the $Q^*$ and $T_{y,y'}$ is estimated correctly the empirical risk of the designed algorithm trained with adversary-aware partial label will converge to the expected risk of the optimal classifier trained with the true label. If the number of sample is reaching infinitely large that given the adversary-aware partial labels, $\hat{f}_n$ is going to converged to $\hat{f}^*$ theoretically. Subsequently, $\hat{f}_n$ will converge to the optimal classifier $f^*$ as claimed in the theorem 1. With the new generation procedure, the loss function risk consistency theorems are introduced.

**Theorem 2.** *The adversary-aware loss function proposed is risk consistent estimator if it can asymptotically converge to the expected risk given sufficiently good approximate of $\bar{Q}$ and the adversary-aware matrix.The proof is in appendix lemma 4.*

$$\mathcal{L}(y, f(x)) = \sum_{\vec{y} \in \vec{\mathcal{Y}}_y} \sum_{y=1}^{C} \sum_{y' \in Y'} (\mathrm{P}(Y = y \mid X = x) = \prod_{b' \in \vec{y}} p_{b'} \cdot \prod_{t' \notin \vec{y}} (1 - p_{t'}) \, \bar{T}_{yy'} \vec{\mathcal{L}}(\vec{y}, f(x))) \quad = \vec{\mathcal{L}}(\vec{y}, f(x)).$$

### 3.0.1 GENERALISATION ERROR

*Define $\hat{R}$ and $\hat{R}_{pn}$ as the true risk the empirical risk respectively given the adversary-aware partial label dataset. The empirical loss classifier is obtained as $\hat{f}_{pn} = \arg\min_{f \in \mathcal{F}} \hat{R}_{pn}(f)$. Suppose a set of real hypothesis $\mathcal{F}_{\vec{y}_k}$ with $f_i(X) \in \mathcal{F}, \forall i \in [c]$. Also, assume it's loss function $\vec{\mathcal{L}}(\boldsymbol{f}(X), \vec{Y})$ is L-Lipschitz continuous with respect to $f(X)$ for all $\vec{y}_k \in \vec{\mathcal{Y}}$ and upper-bounded by $M$, i.e., $M = \sup_{x \in \mathcal{X}, f \in \mathcal{F}, y_k \in \vec{Y}} \vec{\mathcal{L}}(f(x), \vec{y}_k)$. The expected Rademacher complexity of $\mathcal{F}_k$ is denoted as $\Re_n(\mathcal{F}_{\vec{y}_k})$* Bartlett & Mendelson (2002)

**Theorem 3.** *For any $\delta > 0$, with probability at least $1 - \delta$,*

$$\hat{R}\left(\hat{f}_{pn}\right) - \hat{R}\left(\hat{f}^{\star}\right) \leq 4\sqrt{2}L \sum_{k=1}^{c} \Re_n\left(\mathcal{F}_{\vec{y}_k}\right) + M\sqrt{\frac{\log \frac{2}{\delta}}{2n}}. \tag{23}$$

As the number of samples reaches to infinity $n \to \infty, \Re_n(\mathcal{F}_{\vec{y}_k}) \to 0$ with a bounded norm. Subsequently, $\bar{R}(\hat{f}) \to \bar{R}\left(\hat{f}^{\star}\right)$ as the number of training data reach to infinitely large. The proof is given in Appendix Theorem 3.

## 4 EXPERIMENTS

**Datasets** We evaluate the proposed method on three benchmarks-CIFAR10, CIFAR100 Krizhevsky et al. (2009), and fine-grained CUB200 Wah et al. (2011) with general partial label and adversary-aware partial label datasets. **Main Empirical Results for CIFAR10.** All the classification accuracy is shown in Table 1. We have compared classification results on CIFAR-10 with previous works Wang et al. (2022); Lv et al. (2020); Wen et al. (2021) using the Adversarial teacher within momentum (ATM). The method has shown consistently superior results in all learning scenarios where $q = \{0.3, 0.5\}$ for the adversary-aware partial label learning. More specifically, the proposed method achieves **8.17%** superior classification performance at a 0.5 partial rate than the previous state of art work Wang et al. (2022). Moreover, our proposed method has achieved comparable results at 0.1 and 0.3 partial rates. The experiments for CIFAR-10 have been repeated four times with four random seeds. **Main Empirical Results for CUB200 and CIFAR100.** The proposed method has shown superior results for the Adversary-Aware Partial Label, especially in more challenging learning tasks like the 0.1 partial rate of the dataset cub200 and CIFAR100, respectively. On the cub200 dataset, we have shown **5.95%** improvement at partial rates 0.1 and 1.281% and 0.37% where the partial rate is at 0.05 and 0.03. On the CIFAR100 dataset, the method has shown **6.06%** and 0.4181%, 0.5414% higher classification margin at partial rate 0.1, 0.05 and 0.03.The experiments have been repeated five times with five random seeds.

Table 1: Benchmark datasets for accuracy comparisons. Superior results are indicated in bold. Our proposed methods have shown comparable results to fully supervised learning and outperform previous methods in a more challenging learning scenario, such as the partial rate at 0.5(CIFAR10) and 0.1(CIFAR100, CUB200). The hyper-parameter $\alpha$ is set to 0.1 for our method. (The symbol $*$ indicates Adversary-Aware partial label dataset).

| Dataset | Method | $q = 0.01$ | $q = 0.05$ | $q = 0.1$ |
|---|---|---|---|---|
| CIFAR100 | (ATM)(Without T)(Our) | **73.43** ±0.11 | 72.63 ±0.27 | **72.35**±0.22 |
| | PiCO | 73.28 ±0.24 | **72.90** ±0.27 | 71.77±0.14 |
| | LWS | 65.78 ±0.02 | 59.56 ±0.33 | 53.53 ±0.08 |
| | PRODEN | 62.60±0.02 | 60.73±0.03 | 56.80±0.29 |
| | Full Supervised | | 73.56 ±0.10 | |

| Dataset | Method | $q^* = 0.03 \pm 0.02$ | $q^* = 0.05 \pm 0.02$ | $q^* = 0.1 \pm 0.02$ |
|---|---|---|---|---|
| CIFAR100 | (ATM)(Our)$^*$ | 73.36 ±0.32 | 72.76 ±0.14 | **54.09** ±**1.88** |
| | PiCO$^*$ | 72.87 ±0.26 | 72.53 ±0.37 | 48.03±3.32 |
| | LWS$^*$ | 46.8±0.06 | 24.82±0.17 | 4.53±0.47 |
| | PRODEN$^*$ | 59.33±0.48 | 41.20±0.27 | 13.44±0.41 |

| Dataset | Method | $q = 0.01$ | $q = 0.05$ | $q = 0.1$ |
|---|---|---|---|---|
| CUB200 | (ATM) (Without T)(Our) | **74.43**±0.876 | **72.30**±0.521 | **66.87**±0.98 |
| | PiCO | 74.11±0.37 | 71.75±0.56 | 66.12±0.99 |
| | LWS | 73.74±0.23 | 39.74±0.47 | 12.30±0.77 |
| | PRODEN | 72.34±0.04 | 62.56±0.10 | 35.89±0.05 |
| | Full Supervised | | 76.02±0.19 | |

| Dataset | Method | $q^* = 0.03\pm0.02$ | $q^* = 0.05 \pm0.02$ | $q^* = 0.1\pm0.02$ |
|---|---|---|---|---|
| CUB200 | (ATM) (Our)$*$ | **72.22**±1.36 | **72.43**±0.86 | **56.26**±0.70 |
| | PiCO$^*$ | 71.85 ±0.53 | 71.15 ±0.41 | 50.31 ±1.01 |
| | LWS$^*$ | 9.6 ±0.62 | 4.02 ±0.03 | 1.44 ±0.06 |
| | PRODEN$^*$ | 18.71±0.45 | 17.63 ±0.89 | 17.99 ±0.62 |

| Dataset | Method | $q = 0.1$ | $q = 0.3$ | $q = 0.5$ |
|---|---|---|---|---|
| CIFAR10 | (ATM)(Without T)(Our) | 93.57±0.16 | 93.17±0.09 | 92.22±0.40 |
| | PiCO | **93.74**±0.24 | **93.25**±0.32 | **92.46**±0.38 |
| | LWS | 90.30 ±0.60 | 88.99 ±1.43 | 86.16 ±0.85 |
| | PRODEN | 90.24±0.32 | 89.38±0.31 | 87.78±0.07 |
| | Full Supervised | | 94.91±0.07 | |

| Dataset | Method | $q^* = 0.1 \pm 0.02$ | $q^* = 0.3 \pm 0.02$ | $q^* = 0.5 \pm 0.02$ |
|---|---|---|---|---|
| CIFAR10 | (ATM) (Our) $^*$ | 93.52 ±0.11 | **92.98**±0.51 | **89.62**±0.79 |
| | PiCO$^*$ | **93.64**±0.24 | 92.85±0.43 | 81.45±0.57 |
| | LWS$^*$ | 87.34±0.87 | 39.9±0.72 | 9.89±0.55 |
| | PRODEN$^*$ | 88.80±0.14 | 81.88±0.51 | 20.32±3.43 |

## 4.1 ABLATION STUDY

Figure 2 shows the experimental result comparisons for CUB200 between the adversary-aware loss function and previous loss function before and after the momentum updating. Given equation 17, the uncertainty of the transition matrix $\bar{Q}$ is reduced, leading to a good initialisation for the positive set selection, which is a warm start and plays a vital role in improving the performance of contrastive learning. After we have a good set of positive samples, the prototype's accuracy is enhanced. Subsequently, leveraging the clustering effect and the high-quality representation power of the positive sample set of contrastive loss function to improve the classification performance.

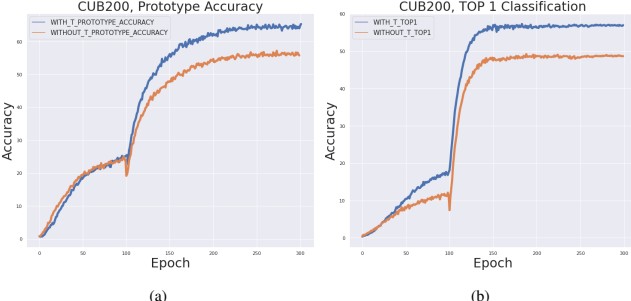

(a)        (b)

Figure 2: The Top1 and Prototype Accuracy of the Proposed Method and the Method in Wang et al. (2022) on CUB200 Adversary-Aware Loss Comparison.

## 5 CONCLUSION AND FUTURE WORKS

This paper introduces a novel Adversary-Aware partial label learning problem. The new problem setting has taken local data privacy protection into account. Specifically, we have added the rival to the partial label candidate set as encryption for the dataset. Nonetheless, the generation process has made the intractable transition matrix even more complicated, leading to an inconsistency issue. Therefore, the novel adversary-aware loss function and the self-attention prototype are proposed. The method is proven to be a provable classifier and has shown superior performance. Future work will use variational inference methods to approximate the intractable transition matrix.

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
