# OpenReview forum: "ADVERSARY-AWARE PARTIAL LABEL LEARNING WITH LABEL DISTILLATION"
_ICLR.cc/2023/Conference — Submitted to ICLR 2023_

### Official Review · Reviewer_o6Rr · 2022-10-15

**Confidence:** 4
**Correctness:** 3
**Technical Novelty And Significance:** 2
**Empirical Novelty And Significance:** 2
**Recommendation:** 3

**Clarity, Quality, Novelty And Reproducibility:**

The paper is generally readable. It seems that the results are reproducible, but I did not check it completely.

**Strength And Weaknesses:**

(+) The proposed new setting is potentially interesting.

(+) The experimental results show that the proposed method is effective.


(-) Although the setting, namely adversary-aware partial label learning, is new, I’m not fully convinced by its practical significance and necessity. In other words, do real-world applications really have such problem in partial label learning? Is it possible to provide a *real* case that falls into the scope of this research?

(-) The definitions of “adversary” and “rival labels” need more explanations. To be specific, the name of the setting is “adversary-aware partial label learning”. However, when I read the paper, I notice that the authors only use a transition matrix to describe its relationship with true label. Therefore, I’m not clear what is the meaning of “adversary” here. In fact, even for the random label noise, one may also use a transition matrix to model such relationship. Note that, for adversary, it means one intends to take some measures to degrade the model performance. However, in the developed model, I cannot see how to resist such attack or the so-called “rival label noise”.

(-) The writing needs significant improvement. Many typos can be found. For example, “accordingly to the Eq.equation 17”. Moreover, the writing quality in the sections of model, theoretical analyses, and experiments, is not as good as that in the introduction.


**Summary Of The Paper:**

This paper presents a novel problem setting named adversary-aware partial label learning. The authors propose an Adversary-Aware loss function and the immature teacher within momentum (ITWM) disambiguation algorithm to tackle this problem.

**Summary Of The Review:**

This paper is potentially interesting. However, it contains some immature factors, especially the practical meaning and the handling of adversary, so I think it is not suitable to be accepted currently.

---

> ### Author Response · Authors · 2022-11-18
> **Reply to Reviewer o6Rr**
>
> $\textbf{Question4}$
>
> The writing needs significant improvement. Many typos can be found. For example, “accordingly to the Eq.equation 17”. Moreover, the writing quality in the sections of model, theoretical analyses, and experiments, is not as good as that in the introduction.
>
> $\textbf{Answer4}$
>
> We have uploaded the new version based on your comments.
>
>
> $\textbf{Reference}$:
>
> Stanley L Warner. Randomized response: A survey technique for eliminating evasive answer bias.
> Journal of the American Statistical Association, 60(309):63–69, 1965.
>
> Peter Kairouz, Sewoong Oh, and Pramod Viswanath. Extremal mechanisms for local differential
> privacy. Advances in neural information processing systems, 27, 2014.
>
> Haobo Wang, Ruixuan Xiao, Yixuan Li, Lei Feng, Gang Niu, Gang Chen, and Junbo Zhao. Pico: Contrastive label disambiguation for partial label learning. ICLR, 2022.

---

> ### Author Response · Authors · 2022-11-19
> **Reply to Reviewer o6Rr**
>
>
>
>
> $\textbf{Question3}$
>
> In fact, even for the random label noise, one may also use a transition matrix to model such relationship. Note that, for adversary, it means one intends to take some measures to degrade the model performance. However, in the developed model, I cannot see how to resist such attack or the so-called “rival label noise”.
>
>
> $\textbf{Answer3}$
>
> We can observe from Table 1 and Figure 2 that the proposed method has shown superior classification performance compared with the previous partial label methods. Most importantly, there are two motivations why introducing rival label. Firstly, it is introduced to address a more challenging and privacy-enhancing learning task. The rival label noise brings controllable randomness to enhance privacy, even if the training data is leaked to the adversary.
>
> In the following, we have shown how the classification performance is impacted if the entries of the class instance-dependent transition matrix $T_{\textbf{Original}}$ is updated to $\bar{T}_{\textbf{New }}$ to show the robustness of our proposed method. We can observe from the table, our method has shown high robustness towards the adversary-aware partial label.
>
> The classification performance comparison for the CIFAR100 dataset with $\bar{T}_{\textbf{New }}$.
>
> $\space$
> DATASET $\space$  | Method  $\space$ |  $\space$ $ q* $=0.1|
>
> $\space$
> CIFAR100$\space$| Wang et al. (2022) PiCO | 20.941(-24.015)%|
>
> $\space$
> CIFAR100 | Our(ATM)| $\textbf{54.156(0.066)}$%|
>
> The classification performance comparison for the CUB200 dataset with $\bar{T}_{\textbf{New }}$.
>
> $\space$
> DATASET $\space$  | Method  $\space$ |  $\space$ $ q* $=0.1|
>
> $\space$
> CUB200$\space$| Wang et al. (2022) PiCO | 21.22(-25.155)%|
>
> $\space$
> CUB200| Our(ATM)| $\textbf{48.62(-7.64)}$%|
>
> We have seen when the $\bar{T}_{new}$  in which each row has five 0.3 label noise rate, is applied to generate the adversary-aware partial label, and our method has shown high tolerance towards the noise. The proposed method has shown 54.156(0.066) % classification performance compared with PiCO 20.941(-24.015)%, whereas the on the cub200, it drops only 7.64% compared with the PICO dropped 25.155 in percentage.
>
>
> For illustration purposes, let's take ten classes as an example. The entries of $\bar{T}$ are shown in the below tables. In our problem setting, we have a number of classes equal to 100 and 200, correspondingly for CIFAR100 and CUB200. Each row has five entries equal to 0.2 in the original $\bar{T}$ and has five entries equal to 0.3 in each row for new  $\bar{T}$.
>
> $\bar{T}_{original}\in R^{C \times C}$ :=
> \\begin{array}{cc}
>  \[ 0.   & 0.2  & 0  & 0.2  & 0.2  & 0.2\]\\\\
>  \[ 0.2  & 0    & 0.2  & 0.2  & 0  & 0.2\]\\\\
>  \[ 0.2  & 0.2  & 0    & 0.2  & 0  & 0.2\]\\\\
>  \[ 0.2  & 0.2  & 0  & 0    & 0.2  & 0.2\]\\\\
>   \[0.2  & 0.2  & 0  &0.2   & 0    & 0.2\]\\\\
>  \[ 0.2  & 0  & 0.2  & 0.2  & 0.2  & 0.  \]
> \\end{array}
>
> $\bar{T}_{new}\in R^{C \times C}$ :=
> \\begin{array}{cc}
>  \[ 0.   & 0.3  & 0  & 0.3  & 0.3  & 0.3\]\\\\
>  \[ 0.3  & 0    & 0.3  & 0.3  & 0  & 0.3\]\\\\
>  \[ 0.3  & 0.3  & 0    & 0.3  & 0  & 0.3\]\\\\
>  \[ 0.3  & 0.3  & 0  & 0    & 0.3  & 0.3\]\\\\
>   \[0.3  & 0.3  & 0  &0.3   & 0    & 0.3\]\\\\
>  \[ 0.3  & 0  & 0.3  & 0.3  & 0.3  & 0.  \]
> \\end{array}

---

> ### Author Response · Authors · 2022-11-19
> **Reply to Reviewer o6Rr**
>
> We sincerely thank for the reviewers' comments and suggestions.
>
> $\textbf{Question1}$
>
> The definitions of “adversary” and “rival labels” need more explanations. To be specific, the name of the setting is “adversary-aware partial label learning”. However, when I read the paper, I notice that the authors only use a transition matrix to describe its relationship with true label. Therefore, I’m not clear what is the meaning of “adversary” here.
>
> $\textbf{Answer1}$
>
> The adversary-aware means our designed paradigm can prevent the adversary from efficiently and reliably inferring certain information from the database without the $\bar{T}$, even if the data was leaked. We hope that we have made both concepts more clear. Our method introduces the rival as controllable randomness to enhance privacy. The rival labels are generated depending on the transition matrix $\bar{T}$. Please refer to section 1.2.2 for detail.
>
> $\textbf{Question2}$
>
> Although the setting, namely adversary-aware partial label learning, is new, I’m not fully convinced by its practical significance and necessity. In other words, do real-world applications really have such problem in partial label learning? Is it possible to provide a real case that falls into the scope of this research?
>
> $\textbf{Answer2}$
>
> Privacy-enhancing techniques in weakly supervised learning should be noticed. Our proposed label generation bears some resemblance to local differential privacy Kairouz et al.(2014); Warner (1965), which aims to randomise the responses from the participants. The potential application is to randomise survey responses, a survey technique for improving the reliability of responses to confidential interviews or private questions. We first collect the correct answer and then randomise the response through some mechanism, in which our case is through $\bar{T}$. In recent days, we have seen a lot of data leak incidents worldwide which had huge implications for personal privacy. The importance is significant and needs to be addressed urgently. The proposed technique can also be applied to other partial-label learning tasks.

---

### Official Review · Reviewer_8DLJ · 2022-10-21

**Confidence:** 5
**Correctness:** 1
**Technical Novelty And Significance:** 2
**Empirical Novelty And Significance:** 1
**Recommendation:** 3

**Clarity, Quality, Novelty And Reproducibility:**

The paper is poorly written, lacks novelty, and the experiments are unconvincing.

**Strength And Weaknesses:**

Strengths:
It is difficult for me to tell the strengths about this paper because the extremely confusing notation system and language makes it difficult for readers to understand.

Weaknesses:
1. The paper is poorly written.
Firstly, the paper does not have a formal definition of the adversary-aware partial label learning problem. From Eq2, the paper seems to introduce a hidden variable $Y'$ to generate the set of candidate labels $\vec{Y}$, but in the definition on page 4, the generation of $\vec{Y}$ is only related to the real label $Y$, which is contradictory. I am really confused about how the setting of the paper differs from the classic partial label learning.
Secondly, most of the notations are not explained. For example, the notions that first appear throughout Sec1.3, $F, \boldsymbol{w}, \boldsymbol{v}, \boldsymbol{u}$ in Sec2.1, etc.
In addition, the methodological and theoretical parts of the article are also very vague. Assumption2 and Lemma2 appear out of thin air on page 7.
2. The methodology section appears to be almost an incremental improvement on PiCo, the only difference being the use of prototypes instead of queues. My bigger confusion is that the proposed method seems to be disconnected from the previous setting, as it does not use $Q$ and it is unclear how to get $T$.
3. The experiment is not convincing. There is no explanation of how to generate the settings studied in the paper, i.e., adversary-aware partial labels, and the reproducibility of the algorithm is poor.

**Summary Of The Paper:**

This paper attempts to proposed a new method for so-called adversary-aware partial label learning (but the definition of this setting is not clear in the whole paper). The proposed method is an incremental improvement on a SOTA partial label learning method.

**Summary Of The Review:**

In its current state, the paper is not ready for publication.

---

> ### Author Response · Authors · 2022-11-19
> **Relpy to Reviewer 8DLJ**
>
> We are sincerely thankful for your valuable comments and helpful suggestions.
>
> $\textbf{Question1}$
>
> The paper is poorly written. Firstly, the paper does not have a formal definition of the adversary-aware partial label learning problem.
>
> $\textbf{Answer1}$
>
> We have updated the new draft for review. Please refer to the section 1.2 of the updated draft.
>
> $\textbf{Question2}$
>
> From Eq2, the paper seems to introduce a hidden variable  to generate the set of candidate labels , but in the definition on page 4, the generation of is only related to the real label , which is contradictory.
>
> $\textbf{Answer2}$
>
> We sincerely thank the review effort for reviewing our draft. However, the review may have some misunderstanding of part of our paper. The generation of adversary partial label depends on both $Y^{\prime}$ and Y which is already stated in Equation 3. The conditional probability of $\vec{Y}$ under the condition $Y^{\prime}=y^{\prime}$ and $Y=y$ is given in Equation 5, 6. Please refer to page 4 of our updated submitted draft.
> % The new learnability condition has been stated in Equation 1.
>
> $\textbf{Question3}$
>
> I am really confused about how the setting of the paper differs from the classic partial label learning.
>
> $\textbf{Answer3}$
>
> 1. The difference is shown in the label generation process which is described in section 1.2 of updated draft.
>
> 2. In a nutshell, in the standard partial label learning problem setting, the label generation is based on $\textbf{Equation(2)}$, whereas our label generation is according to $\textbf{Equation(3)}$. Our work introduces the additional rival variable via transition matrix $\bar{T}$.
>
> $\textbf{Question4}$
>
> Secondly, most of the notations are not explained. For example, the notions that first appear throughout Sec1.3,  in Sec2.1, etc. In addition, the methodological and theoretical parts of the article are also very vague. Assumption2 and Lemma2 appear out of thin air on page 7.
>
> $\textbf{Answer4}$
>
> There is a typo. Assumption 2 is supposed to be assumption 1. We are not sure we have given Lemma2. We have uploaded the new version based on your comments.
>
> $\textbf{Question5}$
>
> The methodology section appears to be almost an incremental improvement on PiCo, the only difference being the use of prototypes instead of queues. My bigger confusion is that the proposed method seems to be disconnected from the previous setting, as it does not use  and it is unclear how to get T.
>
> $\textbf{Answer5}$
>
> Our work introduces the rival variable as an privacy enhancing technique for the adversary-aware partial label learning.
>
> The $\bar{T}$ can be used to introduce noise. In our case, the rival variable serves as controllable randomness to enhance privacy against the potential adversary and information leakage. In contrast, the previous methods can not guarantee privacy protection property.
>
> In reciprocal, it also has the property to remove noise. For instance, equation 17 is designed to help rectify the misclassified prediction by incorporating the $\bar{T}$. Given the debiased prediction, the precision of the prototype is also enhanced. Subsequently, we can obtain a more accurate positive sample set for contrastive loss. Contrastive learning leverages the high quality of representation power to improve the robustness against the rival. Thus, by using equations 17 and 18, we can make the classifier more robust.
>
> The rival is generated manually for privacy enhancement, thus the $\bar{T}$ is given by design.
>
> Please also refer to Reviewer o6Rr Answer 3. We have conducted an experiment to verify our statement.
>
> $\textbf{Question 6}$
>
> The experiment is not convincing. There is no explanation of how to generate the settings studied in the paper, i.e., adversary-aware partial labels, and the reproducibility of the algorithm is poor.
>
> $\textbf{Answer 6}$
>
> Please refer to supplementary page. We have updated the algorithm table.
>
> Figure 2 shows the experimental result comparisons for CUB200 between the adversary-
> aware loss function and previous loss function Wang et al. (2022) before and after the
> momentum updating. Given equation 17, the uncertainty of the transition matrix $Q*$ is reduced, leading to a good initialisation for the positive set selection, which plays a vital role in improving the performance of contrastive learning. After we have a good set of positive samples, the prototype's accuracy is enhanced. Subsequently, leveraging the clustering effect and the high-quality representation power of the contrastive loss function to improve the classification performance.
>
>
> $\textbf{Reference}:$
> Haobo Wang, Ruixuan Xiao, Yixuan Li, Lei Feng, Gang Niu, Gang Chen, and Junbo Zhao. Pico: Contrastive label disambiguation for partial label learning. ICLR, 2022.

---

> > ### Comment · Reviewer_8DLJ · 2022-12-10
> > **Thanks for the response**
> >
> > I appreciate your hard work and the rebuttal. However, I still think there are major problems with the writing and organization of this paper as mentioned by other reviewers, even though the authors almost rewrote Sec. 1. For example, "Lemma 1 is the new ERM learnability condition" comes out of nowhere in page 3. Eq. 7 is wrong given the definition of Q before.
> >
> > More importantly, I couldn't even tell what problem the author is trying to solve with this paper, what are the main points and contributions. The author claims that "we propose an affordable and practical approach to manually corrupt the collected dataset to prevent the adversary from obtaining high-quality, confidential information meanwhile ensure the trustee has full access to the useful information". This goal appears to be achieved by first corrupting the true labels before generating the partial labels, where the trustee has access to the process of corrupting the true label and the attacker does not. What the readers would then naturally like to see is a justification for the idea that how this access right affects the learnability of the adversary-aware PLL problem (by the way, as far as I know this paper may be not the first to propose this problem [lv2021]). Yet the rest of the paper does not seem to address this issue at all, but instead proposes (from the trustees' perspective) a PLL approach, which does not support the authors' claims. So the contribution of this paper is to propose a adversary-aware PLL method, not to do with privacy protection? But given the writing so far, I can't tell if the algorithm is convincing either.
> >
> > Therefore, I regret to say I will increase my score to 3.
> >
> > [lv2021] J. Lv, et al. On the robustness of average losses for partial-label learning.

---

### Official Review · Reviewer_reQY · 2022-10-25

**Confidence:** 2
**Correctness:** 2
**Technical Novelty And Significance:** 2
**Empirical Novelty And Significance:** 2
**Recommendation:** 3

**Clarity, Quality, Novelty And Reproducibility:**

Though the paper seems to introduce a novel and somewhat significant research problem, the paper in its current form has severe issues with the presentation of the proposed problem/method.

**Strength And Weaknesses:**

Strengths:

1. The problem setting is a new interesting direction, and a solution is also presented to pave way to a new research area, with potentially profound implications in privacy enhancing PLL.

Weaknesses / Major Concerns:

1. My primary concern is the presentation of the paper. The current writing is very convoluted with too many concepts, without being proper introduced first. For example, the authors talk about the problem setting in Sec 1 (Page 2), but many notations were not introduced until Sec 1.2 (Page 3). Moving on, some challenges with respect to the proposed problem setting are discussed on Page 2, but it is very unclear and hard to follow (see point 2 below). With all the doubts (as a reader who hasn't read every single reference), the problem is set in Sec 1.2 (Page 3) without proper justifications (see point 3 below).
2. Given the current writing, I find it very hard to understand the motivation of the work when reading through Sec 1 (Page 2). To give a few examples, why is the assumption that the rival is generated depending only on Y but instance X (e.g. simplification for theoretical analysis, or intractability, or practicality)? Why does the inclusion of rival imply an inconsistent classifier according to Eq 2? Then why one cannot obtain consistent classifier due to intractability of \bar{Q}? How does the proposed ITWM help approximate \bar{Q} hence the consistency issue? Are there other alternatives one may consider, and why ITWM in particular?
3. The two main parts of the paper adversary aware PLL (Sec 1.2) and the ITWM algorithm (Sec 2.1) both lack clear explanation. For example, why the choice of the particular form Eq 4 for this problem (e.g. additive instance dependent noise but class dependent transition of rival)? How does the proposed setting extend and differ from Wen et al (2021) (e.g. a dedicated and elaborate section to such related works, or at least in appendix, is much appreciated)? How does instance embedding ("prototype") and the introduction of sparse matrix (A) help instead of obtaining true transition matrix (Page 4)? Why the positive sample set which appeared all of a sudden in Sec 1.3 without any mention throughout? In Sec 2.1, ITWM introduced a bunch of new notations that seem very disconnected from the previous texts? Then why does the adversary aware loss combine two terms (Eq 20) and how is the contrastive loss needed?
4. With all the doubts above, I cannot properly judge the theoretical analysis (Sec 3) or the experiments (Sec 4).

**Summary Of The Paper:**

The authors proposed a novel problem setting of adversary aware partial label learning (PLL) and a novel solution including an adversary aware loss and immature teacher within momentum (ITWM) to solve it. Theoretical analysis are presented and some empirical results are reported.

**Summary Of The Review:**

My major concern for the paper in its current form is the writing. Without properly rewriting the entire paper, I can hardly recommend a higher score. If the authors are willing to polish the writing significantly during rebuttal, I'm happy to have another read and adjust my recommendation accordingly.

---

> ### Author Response · Authors · 2022-11-19
> **Reply to  Reviewer reQY**
>
>
>
> We are sincerely thankful for your valuable comments and helpful suggestions.
>
> $\textbf{Question 1}$
>
> My primary concern is the presentation of the paper. The current writing is very convoluted with too many concepts, without being proper introduced first. For example, the authors talk about the problem setting in Sec 1 (Page 2), but many notations were not introduced until Sec 1.2 (Page 3). Moving on, some challenges with respect to the proposed problem setting are discussed on Page 2, but it is very unclear and hard to follow (see point 2 below). With all the doubts (as a reader who hasn't read every single reference), the problem is set in Sec 1.2 (Page 3) without proper justifications (see point 3 below).
>
> $\textbf{Answer 1}$
>
> We have reorganised the draft and uploaded new version.
>
> $\textbf{Question 2}$
>
> Given the current writing, I find it very hard to understand the motivation of the work when reading through Sec 1 (Page 2).
>
> $\textbf{Answer 2}$
>
> Thanks for your constructive suggestions. We have reorganised the draft and uploaded new version. We hope the following answers can clarify the motivation a bit more based on your comments.
>
> $\textbf{Question 3}$
>
> why is the assumption that the rival is generated depending only on Y but instance X (e.g. simplification for theoretical analysis, or intractability, or practicality)?
>
> $\textbf{Answer 3}$
>
> If the rival is only dependent on Y, the label noise transition matrix is simplified and mathematically identifiable.
> Since all the instances share the same label noise transition matrix in practice, such encryption is more affordable to implement.
>
> $\textbf{Question 4}$
>
> Why does the inclusion of rival imply an inconsistent classifier according to Eq 2? Then why one cannot obtain consistent classifier due to intractability of $\bar{Q}$? How does the proposed ITWM help approximate $\bar{Q}$ hence the consistency issue?
>
> $\textbf{Answer 4}$
>
> 1. We have redefined $\bar{Q}$ as $Q^{*}$ in the updated draft.
>
> 2. Without the information of transition matrix $\bar{T}$, the $Q^{*}$ is intractable to identify, thus we can not achieve an consistent classifier.
>
> 3. If the $\bar{T}$ is given, the uncertainty of the transition matrix $Q^{*}$ is reduced. In practice, the $\bar{T}$ helps us to gain a good initialisation for the positive set selection, meaning the accuracy of it is sufficiently high( Please refer to section 1.3 of the updated draft). It plays an essential role in improving the performance of contrastive learning. After we have produced a good set of positive sample, the precision of the prototype is also enhanced, in turn, leveraging the clustering effect of the contrastive loss function. Figure 2 shows given $\bar{T}$, the accuracy of prototype $v$ and classification performance have significantly improved.
>
> 4. More specifically, the ATM is proposed to identify the Q* by using the pseudo label according to \textbf{equation 17}. It implies the high precision of $v$ is indispensable in identifying the Q* . Our loss function is proposed to improve the precision of $v$ by increasing the accuracy of the positive sample selection, in turn, better label disambiguation (Neural network classification performance). The $v$ is the prototype embedding with $R^{d \times k}$. The $v_{y}$ is a prototype embedding for the y-$th$ class of an instance. We define the $u$ and $k$ as the query and key latent feature from the feature extraction network $f_{\Theta}$  and  $f_{\Theta}^{\prime}$ respectively. We have the output $\boldsymbol{u}\in R^{1 \times d}$ in which $u_{i} = f_{\Theta}(Aug_{q}(x))$ and $z\in R^{1 \times d}$ where $z_i$ $=f_{\Theta}^{\prime}$ $(Aug_{k}(x_{i}))$.The $\textbf{Figure 2}$ shows using our proposed adversary-aware loss function, the prototype accuracy has significantly improved, and in turn, better classification accuracy is acquired.
>
>
> $\textbf{Question 5}$
>
> Are there other alternatives one may consider, and why ITWM in particular?
>
> $\textbf{Answer 5}$
>
> Our proposed paradigm and loss function can be applied universally to other related partial-label learning methods to enhance privacy protection. In contrast, the previous methods can not guarantee privacy protection property.

---

> > ### Author Response · Authors · 2022-11-19
> > **Reply to Reviewer reQY**
> >
> >
> > $\textbf{Question 6}$
> >
> > why the choice of the particular form Eq 4 for this problem (e.g. additive instance dependent noise but class dependent transition of rival)?
> >
> > $\textbf{Answer 6}$:
> >
> > The instance-dependent partial label family in equation 4 is broader than the previous works, while this particular form is feasible in practice.
> >
> > If the $\bar{T}$ is given, the uncertainty of the transition matrix $\bar{Q}$ is reduced. In practice, the $\bar{T}$ helps us to gain a good initialisation for the positive set selection, meaning the accuracy of it is sufficiently high . It plays an essential role in improving the performance of contrastive learning. After we have a good set of positive samples, the precision of the prototype is enhanced, in turn, leveraging the clustering effect of the contrastive loss function. Figure 2 shows shown given $\bar{T}$, the accuracy of prototype v and classification performance have significantly improved simultaneously.
> >
> > $\textbf{Question 7}$
> >
> > How does the proposed setting extend and differ from Wen et al (2021) (e.g. a dedicated and elaborate section to such related works, or at least in appendix, is much appreciated)?
> >
> > $\textbf{Answer 7}$:
> >
> > In the work of Wen et al(2021), label generation is a standard setting in the literature. In contrast, our work introduces the additional rival variable via transition matrix $\bar{T}$. Furthermore, depending on the sophistication of the adversary, our method offers a dynamic mechanism for privacy encryption that is more resilient and flexible to face the potential adversary or privacy risk. By learning from the previous attacks, we can design different levels of protection. The levels range from the most stringent to the least stringent. If the adversary is less sophisticated, the partial label generation is adequate against sabotage. In the situation where we have a more sophisticated adversary, by adjusting the $\bar{T}$, we could design different levels of stringent protection. In addition, our frameworks induce optimal trade-offs between the classification performance and security level depending on the attack frequency and strategy by adjusting the class instance-dependent transition matrix.
> >
> > In the following, we have shown how the classification performance is impacted if the entries of the class instance-dependent transition matrix $T_{\textbf{original}}$  is updated to $\bar{T}_{\textbf{new }}$ to show the robustness of our proposed method. We can observe from the table, our method has shown robustness towards the adversary-aware partial label.
> >
> > The classification performance comparison for the CIFAR100 dataset with $\bar{T}_{\textbf{New }}$.
> >
> > $\space$
> > DATASET $\space$  | Method  $\space$ |  $\space$ $ q* $=0.1|
> >
> > $\space$
> > CIFAR100$\space$| Wang et al. (2022) PiCO | 20.941(-24.015)%|
> >
> > $\space$
> > CIFAR100 | Our(ATM)| $\textbf{54.156(0.066)}$%|
> >
> > The classification performance comparison for the CUB200 dataset with $\bar{T}_{\textbf{New }}$.
> >
> > $\space$
> > DATASET $\space$  | Method  $\space$ |  $\space$ $ q* $=0.1|
> >
> > $\space$
> > CUB200$\space$| Wang et al. (2022) PiCO | 21.22(-25.155)%|
> >
> > $\space$
> > CUB200| Our(ATM)| $\textbf{48.62(-7.64)}$%|
> >
> > We have seen when the $\bar{T}_{new}$ is applied to generate the adversary-aware partial label, and our method has shown high tolerance towards the noise. The proposed method has shown 54.156(0.066) % classification performance compared with PiCO 20.941(-24.015)%, whereas the on the cub200, it drops only 7.64% compared with the PICO dropped 25.155 in percentage.
> >
> > For illustration purposes, let's take ten classes as an example. The entries of $\bar{T}$ are shown in the below tables. In our problem setting, we have a number of classes equal to 100 and 200, correspondingly for CIFAR100 and CUB200. Each row has five entries equal to 0.2 in the original $\bar{T}$ and has five entries equal to 0.3 in each row for new  $\bar{T}$.
> >
> >
> > $\bar{T}_{original}\in R^{C \times C}$ :=
> > \\begin{array}{cc}
> >  \[ 0.   & 0.2  & 0  & 0.2  & 0.2  & 0.2\]\\\\
> >  \[ 0.2  & 0    & 0.2  & 0.2  & 0  & 0.2\]\\\\
> >  \[ 0.2  & 0.2  & 0    & 0.2  & 0  & 0.2\]\\\\
> >  \[ 0.2  & 0.2  & 0  & 0    & 0.2  & 0.2\]\\\\
> >   \[0.2  & 0.2  & 0  &0.2   & 0    & 0.2\]\\\\
> >  \[ 0.2  & 0  & 0.2  & 0.2  & 0.2  & 0.  \]
> > \\end{array}
> >
> > $\bar{T}_{new}\in R^{C \times C}$ :=
> > \\begin{array}{cc}
> >  \[ 0.   & 0.3  & 0  & 0.3  & 0.3  & 0.3\]\\\\
> >  \[ 0.3  & 0    & 0.3  & 0.3  & 0  & 0.3\]\\\\
> >  \[ 0.3  & 0.3  & 0    & 0.3  & 0  & 0.3\]\\\\
> >  \[ 0.3  & 0.3  & 0  & 0    & 0.3  & 0.3\]\\\\
> >   \[0.3  & 0.3  & 0  &0.3   & 0    & 0.3\]\\\\
> >  \[ 0.3  & 0  & 0.3  & 0.3  & 0.3  & 0.  \]
> > \\end{array}

---

> > > ### Author Response · Authors · 2022-11-19
> > > **Reply to Reviewer reQY**
> > >
> > > $\textbf{Question 8}$
> > >
> > > How does instance embedding ("prototype") and the introduction of sparse matrix (A) help instead of obtaining true transition matrix (Page 4)?
> > >
> > > $\textbf{Answer 8}$
> > >
> > > We can not achieve the true transition matrix $\bar{Q}$ due to the nature of the practical partial label problem. Therefore, we have used prototype learning to identify the true transition matrix $\bar{Q}$.
> > >
> > > $\textbf{Question 9}$
> > >
> > > Why the positive sample set which appeared all of a sudden in Sec 1.3 without any mention throughout?
> > >
> > > $\textbf{Answer 9}$
> > >
> > > We have given the explanation in the section 1.3 of the updated draft.
> > >
> > > $\textbf{Question 10}$
> > >
> > > In Sec 2.1, ITWM introduced a bunch of new notations that seem very disconnected from the previous texts?
> > >
> > > $\textbf{Answer 10}$
> > >
> > > Sorry. We have uploaded the new version for review. Please refer to section 2.1 of the update draft.
> > >
> > > $\textbf{Question 11}$
> > >
> > > Then why does the adversary aware loss combine two terms (Eq 20) and how is the contrastive loss needed?
> > >
> > > $\textbf{Answer 11}$
> > >
> > > There are two terms of the proposed loss function equation 19, which are the equation 17 and equation 18 correspondingly. The motivation and details are explained below. The contrastive loss is designed to learn high quality representation. The new loss equation 17 is designed as regularisation to prevent the contrastive loss from collapsing due to the introduced rival.
> > >
> > > The quality of representation learning by contrastive learning equation 18 underpins the label disambiguation learning objective. In addition, the representation learning of CL depends on the prediction accuracy for the positive sample set. However, the prediction accuracy of the positive sample set in the prior work is observed to deteriorate drastically due to the introduced rival. Our proposed loss function equation 17 is designed to impede the deterioration of the clustering effect in the embedding space in contrastive loss equation 18. The functionality $\bar{T}$ of the proposed adversary-aware loss function acts as an intermediary (See Figure 1) to modify the misclassified predictions classifier. The reason is that it shares some of the information of the transition matrix $Q*$. Thus the uncertainty from the intractable $Q*$ is reduced. Subsequently, it helps the classifier to learn a more accurate positive sample set; in turn, the precision of the prototype is increased. If the prototype $v_{j}$ is correctly assigned with its corresponding j $\in \{1,..., C\}$, the clustering effect in the latent space of CL will be leveraged, meaning high-quality representation is obtained.
> > >
> > > $\textbf{Reference}$:
> > >
> > > Haobo Wang, Ruixuan Xiao, Yixuan Li, Lei Feng, Gang Niu, Gang Chen, and Junbo Zhao. Pico:
> > > Contrastive label disambiguation for partial label learning. ICLR, 2022.
> > >
> > > Hongwei Wen, Jingyi Cui, Hanyuan Hang, Jiabin Liu, Yisen Wang, and Zhouchen Lin. Leveraged
> > > weighted loss for partial label learning. In Marina Meila and Tong Zhang (eds.), Proceedings of
> > > the 38th International Conference on Machine Learning, volume 139 of Proceedings of Machine
> > > Learning Research, pp. 11091–11100. PMLR, 18–24 Jul 2021

---

### Official Review · Reviewer_ctfM · 2022-10-26

**Confidence:** 3
**Clarity, Quality, Novelty And Reproducibility:** The clarity, quality and originality …
**Correctness:** 3
**Technical Novelty And Significance:** 3
**Empirical Novelty And Significance:** 3
**Recommendation:** 5

**Strength And Weaknesses:**

Pros:
* This paper proposes to take the data privacy into account in partial label learning, which is novel and significant in practice.

* The proposed method is smart and easy to follow, referring the idea of self-supervised learning.

Cons:
* The theoretical analysis is based on the fully rank transition matrix. I have a concern about the existence of this condition especially when the noise is very regular.

* I think the rival is essentially a noisy label for PLL. Thus, incorporating the label noise learning methods into the existing PLL methods will be an important baseline.

* Some typos. For example, in the caption of Figure 2, "the Method inWang et al. (2022)".



**Summary Of The Paper:**

This paper introduces a the adversary-aware partial label learning problem to protect the data privacy. The novel adversary-aware loss function, together with an immature teacher within momentum disambiguation algorithm, has achieved state of-the-art performance and proven to be a provable classifier.

**Summary Of The Review:**

Please refer to the Strength And Weaknesses.

---

> ### Author Response · Authors · 2022-11-17
> **Reply to Reviewer ctFM**
>
> $\textbf{Question 1}$
>
> The theoretical analysis is based on the fully rank transition matrix. I have a concern about the existence of this condition especially when the noise is very regular.
>
> $\textbf{Answer 1}$
>
> Thanks for your valuable comments and helpful suggestions. The assumption of the transition matrix is widely utilized and standardized in the literature Patrini et al. (2017); Han et al. (2018).The full rank of the transition matrix has sufficient information to achieve the Bayes classifier from the noise labels. When the full rank condition of the transition matrix is not met, we can still obtain the classifier as the approximation of the Bayes classifier. In practice, we do not need to invert the transition matrix because we only need the matrix product of the transition matrix during the forward pass.
>
>
>
> $\textbf{Question 2}$
>
> I think the rival is essentially a noisy label for PLL. Thus, incorporating the label noise learning methods into the existing PLL methods will be an important baseline.
>
> $\textbf{Answer 2}$
>
> Thanks for your valuable comments and helpful suggestions. Feng et al. (2020) is the most recent method that applied the re-weighting strategy Liu & Tao (2015) to tackle the partial label learning problem. We have compared our method with the method Feng et al. (2020) on the cifar10, cifar100 dataset. The classification accuracy is shown in the below tables. The experimental results have shown that our proposed method has demonstrated superior performance than the Feng et al. (2020) method.
>
> For the adversary-aware partial label at the partial rate $q^{*}$ = {$\{0.03, 0.05, 0.1\}$} we have implemented the following experiments for cifar10 and cifar100.
>
> $\space$
> Dataset  $\space$ | $\space$ Method | $\space$ $ q* $=0.03 |$\space$ $q*$=0.05 |$\space$ $q^{*}$=0.1 |
>
> $\space$
> CIFAR10 $\space$ | Feng et al. (2020)CC | 79.54 ±0.55%| 76.46 ±0.81%| 25.03±0.37%|
>
> $\space$
> CIFAR100 | Feng et al. (2020)CC | 42.59 ±0.13% | 35.72±0.74% | 5.57±0.27%|
>
> For the partial label at the partial rate $q$ = {$ \{0.01, 0.05, 0.1\}$} we have implemented the following experiments for cifar10 and cifar100.
>
> $\space$
> Dataset  $\space$ | $\space$ Method | $\space$ $ q $=0.01 |$\space$ $q$=0.05 |$\space$ q=0.1 |
>
> $\space$
> CIFAR10 $\space$ | Feng et al. (2020)CC | 82.30% ±0.21 | 79.08±0.07% | 74.05±0.35%|
>
> $\space$
> CIFAR100 | Feng et al. (2020)CC | 49.76% ±0.45 | 47.62±0.08% | 35.72±0.47%|
>
>
> $\textbf{Question 3}$
>
> Some typos. For example, in the caption of Figure 2, "the Method inWang et al. (2022)".
>
> $\textbf{Answer 3}$
>
> Thanks for your valuable comments and helpful suggestions. We have corrected the typos. We have also reorganized the draft and uploaded a new version.
>
>
> $\textbf{Reference}$:
>
> Lei Feng, Jiaqi Lv, Bo Han, Miao Xu, Gang Niu, Xin Geng, Bo An, and Masashi Sugiyama. Provably consistent partial-label learning. Advances in Neural Information Processing Systems, 33:
> 10948–10960, 2020.
>
> Tongliang Liu and Dacheng Tao. Classification with noisy labels by importance reweighting. IEEE
> Transactions on pattern analysis and machine intelligence, 38(3):447–461, 2015.
>
> Bo Han, Jiangchao Yao, Gang Niu, Mingyuan Zhou, Ivor Tsang, Ya Zhang, and Masashi Sugiyama.
> Masking: A new perspective of noisy supervision. Advances in neural information processing systems, 31, 2018
>
> Giorgio Patrini, Alessandro Rozza, Aditya Krishna Menon, Richard Nock, and Lizhen Qu. Making
> deep neural networks robust to label noise: A loss correction approach.In Proceedings of the IEEE conference on computer vision and pattern recognition, pp. 1944–1952, 2017.

---

### Decision · Program_Chairs · 2023-01-20

**Decision:**

Reject

**Justification For Why Not Higher Score:**

see above

**Justification For Why Not Lower Score:**

see above

**Metareview: Summary, Strengths And Weaknesses:**

Unfortunately, the reviewers were not enthusiastic enough about this paper for it to be considered for acceptance at ICLR 2023. There are just too many papers that reviewers were much more enthusiastic about this year, and ICLR has a very low acceptance rate. The authors are encouraged to take the reviewer comments very seriously, even if there are things you disagree with, and make sure that all issues are addressed, and any potential sources of confusion are completely eliminated, in the next version of the paper. Also, please ensure that these are addressed in the initial paper submission for that next conference.

**Summary Of Ac-Reviewer Meeting:**

see above